# OpenReview forum: "AICrypto: Evaluating Cryptography Capabilities of Large Language Models"
_ICML.cc/2026/Conference — ICML 2026 regular_

### Official Review · Reviewer_Qf3U · 2026-03-01

**Soundness:** 2
**Presentation:** 3
**Significance:** 4
**Originality:** 3
**Overall Recommendation:** 4
**Confidence:** 4

**Summary:**

This paper introduces AICrypto, a cryptography-focused benchmark for large language models spanning concept recall, practical vulnerability exploitation, and formal proof reasoning. The benchmark contains 135 rewritten/verified MCQs, 150 CTF crypto challenges standardized into a unified file structure (including helper scripts and a sandboxed execution setup), and 30 proof problems drawn from unreleased course exams/homeworks with detailed rubrics and reference solutions.

The work evaluates 17 contemporary models, comparing them to human expert baselines (MCQs via open-book exam, CTF via top competition scoreboard proxies on a 100-challenge subset, proofs via top scores on the original course materials). The headline finding is that top models can match or exceed human experts on MCQs and get close on proof scoring, but still lag substantially on CTF challenges, with failure modes tied to arithmetic mistakes, shallow pattern matching, and weak multi-step/dynamic reasoning.

**Compliance With Llm Reviewing Policy:**

Affirmed.

**Final Justification:**

Authors addresses my concerns, so I will keep my positive score, and increase significance score from 3 to 4.

**Key Questions For Authors:**

See Weaknesses.

**Limitations:**

Yes.

**Strengths And Weaknesses:**

Strengths:

1) Clear high-level structure: benchmark definition -> task construction -> evaluation protocol -> results and failure analysis; many concrete examples and figures (e.g., example CTF trajectory, proof rubric example) help readers understand what is being evaluated.

2) Well-scoped benchmark design across three modalities (MCQ / CTF / proofs) that meaningfully separates memorization, applied exploitation, and formal reasoning.

3) Agentic CTF evaluation is concrete and reproducible in spirit: explicit action loop, turn cap, pass@3 metric, restricted environment, standardized file structure, and helper scripts to mitigate context-window truncation.

4) Automated proof scoring is thoughtfully implemented: expert-written rubrics/reference solutions and a grader ensemble with multiple samples; correlation with human scores is reported and reasonably high, suggesting the grading signal is meaningful.


Weaknesses:

1) Agentic evaluation frameworks for CTF-style tasks exist in adjacent benchmarks; novelty here is mostly domain specialization + curation quality rather than a fundamentally new evaluation method.

2) The MCQ component appears near-saturated for top models, which reduces its long-term value unless refreshed or made harder (the paper also acknowledges this).

3) The benchmark’s ultimate impact depends heavily on release quality and maintenance (versioning, anti-contamination strategy over time), which isn’t fully discussed. Also, there is risk of benchmark leakage / contamination remains despite efforts (e.g., "2023 or later" sourcing for most CTFs). For MCQs, sources include public sites and university materials that might have partial overlap with training corpora; the paper says questions are rewritten and values modified, but it’s unclear how much that reduces semantic leakage versus surface-form leakage. Authors must add at least one concrete probe: e.g., n-gram overlap or retrieval-based nearest-neighbor checks against public sources, or a “closed-book vs open-book” delta on MCQs, or measuring whether models can “quote” sources.

4)  Since proof evaluation relies on LLM graders, consider adding: inter-grader agreement breakdowns (not just correlation), sensitivity to prompt, and a small “adversarial” set (slightly wrong proofs) to quantify false positives.

---

> ### Author Rebuttal · Authors · 2026-03-27
>
> We really appreciate your time and insightful comments!  We hope the following experiments and discussions can address your questions and concerns. Due to space limitations, we present partial results here, and full results are available at https://anonymous.4open.science/r/aicrypto-CE6E/rbt/Qf3U.md.
>
> **W1: Novelty.**
>
> Thank you for this clarifying point. To the best of our knowledge, no prior work has provided a systematic and comprehensive evaluation of LLM cryptographic capabilities, and we believe this benchmark fills an important gap in the field. In addition, the curation process required over 546 expert hours:
>
> | Part | # Experts | Time (Hours) |
> |-|-|-|
> | **MCQs** | 6 | 156 |
> | **CTF** | 3 | 300 |
> | **Proof** | 5| 90 |
> | **Total** | --| **546** |
>
> and our benchmark roughly doubles both task coverage and the number of evaluated models compared to prior work:
>
> | Benchmark | # Crypto CTF | # Other Crypto Tasks | # Total | # LLMs |
> |-|-|-|-|-|
> | **AICrypto (Ours)** | **150** | **165** | **315** | **17** |
> | CyBench | 16 | – | 16 | 8 |
> | NYUCTF | 62 | – | 62 | 5 |
> | Shao et al. | 4 | – | 4 | 6 |
> | InterCode-CTF | 15 | – | 15 | 2 |
>
> We hope this work could serve as a useful foundation for future research on LLM applications in cryptography.
>
> **W2: MCQ performance saturation.**
>
> Thank you for raising this point. As noted in our limitations section, since no prior work has evaluated the cryptographic capabilities of LLMs, this study serves as a preliminary exploration intended to inform future research and applications. We acknowledge that the MCQ component approaches saturation for top-performing models. However, we believe the finding that state-of-the-art LLMs have acquired fundamental cryptographic concepts remains a meaningful contribution in its own right. We agree that refreshing or extending the difficulty of the MCQ component is a valuable direction for future work.
>
> **W3: Data contamination.**
>
> Thank you for this constructive suggestion. To address the concern about benchmark contamination, we conduct a retrieval-based contamination probe across all 135 MCQs. For each question, we retrieve the top-10 public candidates using the question text with DuckDuckGo HTML search, then measure normalized n-gram overlap between each MCQ and its nearest retrieved source. The results are as follows:
>
> | Metric | Value |
> | --- | ---: |
> | Number of MCQs | 135 |
> | Median 5-gram cont. | 0.0000 |
> | Mean 5-gram cont. | 0.0722 |
> | Median 7-gram cont. | 0.0000 |
> | Mean 7-gram cont. | 0.0368 |
> | 5-gram cont. > 0.3 | 12 / 135 |
> | 5-gram cont. > 0.5 | 4 / 135 |
> | 7-gram cont. > 0.3 | 6 / 135 |
> | 7-gram cont. > 0.5 | 3 / 135 |
>
> Both median containment scores are 0.0, indicating that verbatim overlap with public sources is minimal for most items. The small subset showing higher overlap (at most 12 out of 135 items at the 0.3 threshold) is consistent with our rewriting and value-modification process, which primarily targets surface-form leakage. We acknowledge that semantic leakage is harder to eliminate.
>
> For CTF challenges, we compare models' success rates (SR) on the 85 problems with public official solutions (w/ OS) against the remaining 65 without (w/o OS):
>
> | Model | SR w OS | SR w/o OS |
> |------|------------------|---------------------|
> | gemini-2.5-pro | 47.06% (40/85) | 67.69% (44/65) |
> | o3-high | 47.06% (40/85) | 63.08% (41/65) |
>
> As shown above, models consistently perform worse on problems with public write-ups, suggesting contamination has not inflated performance. We agree that long-term versioning and anti-contamination strategies are important and will include a dedicated discussion in the revised version.
>
> **W4: LLM graders' robustness.**
>
> Thank you for this suggestion. To assess grader robustness, we test two rephrased prompt variants and substitute one deprecated model. The results are shown below:
>
> | Models | Prompt | Pearson | Spearman
> |------|-------|-------|-------|
> | gpt-5.1 + gemini-3-pro-preview | original | 0.9025 |0.8974
> | gpt-5.1 + gpt-5.4 | original| 0.8942 |0.8821 |
> | gpt-5.1 + gpt-5.4 | variant-1 | 0.8846 | 0.877 |
> |gpt-5.1 + gpt-5.4 |variant-2 |0.8994|0.8891|
>
> The Pearson and Spearman correlations with human expert scores remain consistently high across all variants, with Pearson varying by less than 0.02, suggesting the autograder is not overly sensitive to prompt phrasing or model choice.
> We also compute the inter-grader agreement between the two scoring models directly. The Pearson correlation between their scores is 0.8798, and the Spearman correlation is 0.8957 (both p < 1e-10). The mean absolute score gap is 0.54 points (approximately 10% of the scale), and in 84.31% of cases, the two models agree within one point. These results suggest that our autograder is stable and that the two graders reach consistent judgments in the large majority of cases.
>
> Thank you again for your thoughtful comments, and we look forward to addressing any further questions or concerns.

---

> > ### Author Rebuttal · Reviewer_Qf3U · 2026-04-01
> >
> > I really appreciate the time the authors took to address my concerns!
> > I will keep my positive score, and increase significance score from 3 to 4.
> > I wish the authors the best of luck with their submission!

---

> > > ### Author Response · Authors · 2026-04-01
> > >
> > > Dear Reviewer Qf3U,
> > >
> > > Thank you very much for your kind words and for updating your significance score. We truly appreciate the time and effort you took to review our work, and your comments are extremely constructive and helpful. Your feedback is very encouraging to us!
> > >
> > > Best regards,
> > >
> > > Authors

---

### Official Review · Reviewer_aLSb · 2026-03-07

**Soundness:** 3
**Presentation:** 3
**Significance:** 3
**Originality:** 3
**Overall Recommendation:** 4
**Confidence:** 4

**Summary:**

This paper introduces AICrypto, a cryptography benchmark for evaluating LLMs across 135 multiple-choice questions targeting conceptual knowledge, 150 CTF challenges targeting practical vulnerability exploitation, and 30 proof problems targeting formal reasoning. The benchmark is constructed and reviewed by cryptography experts, and proof problems are paired with scoring rubrics plus an automated grading approach that is validated via high correlation with human expert scores. The authors evaluate 17 contemporary models and report a consistent pattern which is that top models achieve very strong results on MCQs and relatively strong performance on proof problems, while performance on CTF challenges remains substantially lower than human experts.

**Compliance With Llm Reviewing Policy:**

Affirmed.

**Final Justification:**

Overall, despite some remaining questions about coverage and grading robustness, the benchmark is carefully constructed, practically valuable, and likely to become a useful resource for the community, so I support weak acceptance.

**Key Questions For Authors:**

1. Beyond recent CTFs and rewritten MCQs, what steps were taken to limit training data contamination for the evaluated models, especially for well-known public CTFs and widely circulated exam materials? How would the conclusions change under a stricter verifiably uncontaminated subset?

2. You report strong correlation between human and automated grading. Can you break this down by topic category and difficulty level? Are there systematic failure modes where graders overscore stylistically polished but incorrect proofs?

3. Since the CTF evaluation uses an agentic framework with a 100-turn cap and pass@3, how sensitive are results to these design choices? Would rankings shift with stronger scaffolding?

4. How should readers interpret the benchmark’s coverage relative to modern cryptography practice which includes protocol level reasoning, formal methods, implementation hardening? What is the intended boundary of cryptography capability measured here?

**Limitations:**

yes

**Strengths And Weaknesses:**

Strengths

The benchmark construction is clearly structured with task taxonomy and expert involvement with the evaluation protocols being specified per task type such as pass@3 for CTF or best-of-three for proof scoring. The automated proof grading is supported by strong agreement with human assessments through Pearson correlations reported on a sizable sample, which improves credibility.

The narrative is easy to follow starting with a benchmark motivation to the benchmark design to the evaluation protocols to the final results breakdown and failure analysis.

Cryptography is an important field, and a dedicated benchmark that spans knowledge, hands-on exploitation, and proof-based reasoning fills a real gap versus broader cybersecurity benchmarks where cryptography is only a minor portion. The finding that CTF performance lags far behind humans seems practically important for both cybersecurity AI applications and security of vibe-coding.

A cryptography specific, multiformat benchmark with expert reviewed items and scalable proof grading is a meaningful synthesis and is made reusable by the community.

Weaknesses

Using LLMs as graders can introduce some bias such as favoring certain proof styles or surface plausibility although the paper mitigates this partially via correlation to humans. Additional robustness checks with calibration across difficulty bands or adversarial plausible but incorrect proofs would strengthen the claim further.

Some of the categorical heatmaps are information dense so short guiding captions could help explain how to interpret the composite score and task weights for non-specialists.

The paper frames itself as cryptography capabilities broadly, but the benchmark is still only one operationalization of that space. For example, protocol design, formal verification workflows, side channel reasoning, and composable security arguments are not clearly represented in the main paper.

Benchmarking LLMs on CTF tasks and on proof tasks has precedent in adjacent communities. The novelty is based on curation quality, cryptography focus, and unified design rather than a new ML method.

---

> ### Author Rebuttal · Authors · 2026-03-27
>
> We really appreciate your time and insightful comments!  We hope the following experiments and discussions can address your questions and concerns. Due to space limitations, we present partial results here, and full results are available at https://anonymous.4open.science/r/aicrypto-CE6E/rbt/aLSb.md.
>
> **W1: LLMs as graders.**
>
> Thank you for this suggestion.  To further assess robustness, we test two rephrased prompt variants and substitute one deprecated model:
>
> | Models | Prompt | P | S
> |-|-|-|-|
> | gpt-5.1 + gemini-3-pro-preview | original | 0.90 |0.89
> | gpt-5.1 + gpt-5.4 | original| 0.89 |0.88
> | gpt-5.1 + gpt-5.4 | variant-1 | 0.88 | 0.87
> |gpt-5.1 + gpt-5.4 |variant-2 |0.89|0.88
>
> The Pearson and Spearman correlations with human expert scores remain consistently high across all variants, varying by less than 0.02. This suggests our autograder is robust and not overly sensitive to prompt phrasing or model choice.
>
> **W2: Short caption.**
>
> Thank you for the suggestion. In the revised version, we will add concise captions to the categorical heatmaps to help readers interpret the composite scores and task weights more easily.
>
> **W3: Other cryptographic capabilities.**
>
> Thank you for this observation. We acknowledge that our benchmark is one operationalization of cryptographic capability and that important areas remain uncovered. As there is little prior work in this space, we view our benchmark as a starting point rather than a definitive framework, and we hope it serves as a foundation for future exploration.
>
> **W4: Novelty.**
>
> Thank you for this clarifying point. We agree that our novelty lies in curation quality, cryptography focus, and unified design rather than a new ML method. We believe that building a rigorous benchmark is a substantial contribution in its own right. The curation process required over 546 expert hours:
>
> | Part | # Experts | Time (Hours) |
> |-|-|-|
> | **MCQs** | 6 | 156 |
> | **CTF** | 3 | 300 |
> | **Proof** | 5| 90 |
> | **Total** | --| **546** |
>
> and our benchmark roughly doubles both task coverage and the number of evaluated models compared to prior work:
>
> | Benchmark | # Crypto CTF | # Other Crypto Tasks | # Total | # LLMs |
> |-|-|-|-|-|
> | **AICrypto (Ours)** | **150** | **165** | **315** | **17** |
> | CyBench | 16 | – | 16 | 8 |
> | NYUCTF | 62 | – | 62 | 5 |
> | Shao et al. | 4 | – | 4 | 6 |
> | InterCode-CTF | 15 | – | 15 | 2 |
>
> **Q1: Data contamination.**
>
> Thank you for raising this important concern.  We address contamination risk separately for each benchmark component. Proof problems are entirely original and have not appeared publicly.
> For MCQs, a retrieval-based n-gram overlap probe across all 135 questions shows that both median containment scores are 0.0, indicating minimal verbatim overlap:
>
> | Metric | Value |
> | - | -: |
> | Median 5-gram cont. | 0.0000 |
> | Mean 5-gram cont. | 0.0722 |
> | Median 7-gram cont. | 0.0000 |
> | Mean 7-gram cont. | 0.0368 |
>
> For CTF challenges, we compare models' success rates (SR) on the 85 problems with public official solutions (w/ OS) against the 65 without (w/o OS). Models consistently perform worse on the former, suggesting contamination has not inflated performance:
>
> | Model | SR w OS | SR w/o OS |
> |------|------------------|---------------------|
> | gemini-2.5-pro | 47.06% (40/85) | 67.69% (44/65) |
> | o3-high | 47.06% (40/85) | 63.08% (41/65) |
> | o3 | 42.35% (36/85) | 58.46% (38/65) |
>
> **Q2: Detailed correlation.**
>
> Thank you for this thoughtful question. We provide a breakdown of the correlation between human and automated grading by topic category in the table below:
>
> | Task Category | # Tasks | Pearson |
> |-|-|-|
> | **All** | 306 | **0.903** |
> | ENC | 119 | 0.878 |
> | FUN | 85 | **0.925** |
> | PR | 102 | 0.874 |
>
> The Pearson correlation between the autograder and human ratings remains consistently around 0.90 across all three topic subcategories, suggesting the automated grader performs reliably regardless of topic.
>
> **Q3: Hyperparameter.**
>
> Thank you for this question. We believe 100 turns and pass@3 already represent fairly lenient conditions. To examine this further, we analyze the average number of iterations for both successful and failed attempts across models, with results shown below:
>
> | Model | Avg Iter (Success) | Avg Iter (Fail) |
> |-|-:|-:|
> | gemini-2.5-pro | 17.76 | 35.84 |
> | gpt-4.1 | 12.97 | 27.43 |
>
> Most models use far fewer than 100 iterations on average, even in failed attempts, suggesting the cap is not a binding constraint. This indicates that stronger scaffolding alone is unlikely to substantially shift model rankings.
>
> **Q4: Interpretation.**
>
> Thank you for this question. Our benchmark covers conceptual understanding, applied problem-solving, and proof-based reasoning as a first step. Protocol-level reasoning and formal methods are important, but beyond this initial scope. We hope future work builds on this foundation.
>
> Thank you again for your thoughtful comments, and we look forward to addressing any remaining concerns or questions.

---

> > ### Author Rebuttal · Reviewer_aLSb · 2026-04-01
> >
> > Thank you so much for taking the time to thoroughly address all concerns. We believe this paper is still a weak accept and strongly disagree with the one reviewer who gave a rejection. We appreciate the time spent on the appendix as well for benchmark transparency.

---

> > > ### Author Response · Authors · 2026-04-01
> > >
> > > Dear Reviewer aLSb,
> > >
> > > We sincerely thank you for your thorough review and for the time you spent examining both the main paper and the appendix. Your detailed and constructive comments are extremely helpful and encouraging. We truly appreciate the effort you put into understanding our work and providing thoughtful feedback!
> > >
> > > Best regards,
> > >
> > > Authors

---

### Official Review · Reviewer_q2tR · 2026-03-13

**Soundness:** 2
**Presentation:** 3
**Significance:** 2
**Originality:** 2
**Overall Recommendation:** 4
**Confidence:** 3

**Summary:**

This paper introduces AICrypto, a benchmark for evaluating LLMs on cryptographic tasks, comprising 2,945 MCQs (18 undergrad topics), 30 proof problems (grad-level), and 150 CTF challenges (from recent competitions). Evaluations on 17 SOTA LLMs show they achieve near-human performance on MCQs but lag significantly on CTFs (best 54% vs. human 81%) and struggle with rigorous proof generation. This reveals that while LLMs excel at knowledge recall, they significantly lag behind human experts in practical exploitation and rigorous proof generation.

**Compliance With Llm Reviewing Policy:**

Affirmed.

**Final Justification:**

The rebuttal clearly addresses my concerns. The authors justify the benchmark contribution with strong coverage and comparison, and provide convincing analysis showing minimal data contamination. Limitations on proof problems and the agent framework are reasonably explained.

Overall, my concerns are resolved, and I maintain my positive score of 4.

**Key Questions For Authors:**

See weaknesses.

**Limitations:**

yes

**Strengths And Weaknesses:**

**Strengths**

1. The paper addresses an important and timely problem: evaluating LLM capabilities in cryptography and security.

2. The benchmark covers multiple dimensions of cryptographic competence, including knowledge, practical exploitation, and formal reasoning.

3. Experiments cover a large set of state-of-the-art LLMs and provide useful insights into current limitations.

---

**Weaknesses**

1. The main contribution is a benchmark and empirical evaluation, with limited methodological novelty.

2. Potential training data contamination cannot be fully ruled out, especially for well-known cryptographic problems.

3. The number of proof problems is relatively small and may not fully capture the diversity of cryptographic reasoning.

4. The CTF evaluation relies on a relatively simple agent framework, which may underestimate model capabilities.

---

> ### Author Rebuttal · Authors · 2026-03-27
>
> We really appreciate your time and insightful comments!  We hope the following experiments and discussions can address your questions and concerns. Due to space limitations, we present partial results here, and full results are available at https://anonymous.4open.science/r/aicrypto-CE6E/rbt/q2tR.md.
>
> **W1: Novelty.**
>
> Thank you for this thoughtful observation. We acknowledge that the primary contribution of this work is the benchmark itself and the accompanying empirical evaluation, rather than methodological novelty. We believe that establishing a comprehensive cryptography benchmark is a meaningful contribution, as it addresses a clear gap in the field and provides a foundation for future research. We have also invested substantial expert effort to ensure the benchmark's quality, as summarized below:
>
> | Part | # Experts | Time (Hours) |
> |-|-|-|
> | **MCQs** | 6 | 156 |
> | **CTF** | 3 | 300 |
> | **Proof** | 5| 90 |
> | **Total** | --| **546** |
>
> Furthermore, our benchmark considerably expands the scope of existing work, roughly doubling both the number of tasks and the number of evaluated models:
>
> | Benchmark | # Crypto CTF | # Other Crypto Tasks | # Total | # LLMs |
> |------|-----------|-------|--------|--------|
> | **AICrypto (Ours)** | **150** | **165** | **315** | **17** |
> | CyBench | 16 | – | 16 | 8 |
> | NYUCTF | 62 | – | 62 | 5 |
> | Shao et al. | 4 | – | 4 | 6 |
> | InterCode-CTF | 15 | – | 15 | 2 |
>
> We hope these points offer some reassurance regarding the value of our contribution.
>
> **W2: Data contamination.**
>
> Thank you for raising this concern. We agree that training data contamination cannot be entirely ruled out, particularly for well-known cryptographic problems. To mitigate this, we take several steps: MCQs are rewritten rather than copied verbatim from existing sources, we prioritize the most recent CTF challenges, and the proof problems are original and have not been publicly released before this work.
>
> To further assess contamination in the MCQ component, we conduct a retrieval-based probe across all 135 questions. For each question, we retrieve the top-10 public candidates via DuckDuckGo search and measure normalized n-gram overlap between each MCQ and its nearest retrieved source. Here are the results:
>
> | Metric | Value |
> | --- | ---: |
> | Median 5-gram cont. | 0.00 |
> | Mean 5-gram cont. | 0.0722 |
> | Median 7-gram cont. | 0.00 |
> | Mean 7-gram cont. | 0.0368 |
> | 5-gram cont. > 0.3 | 12 / 135 |
> | 5-gram cont. > 0.5 | 4 / 135 |
> | 7-gram cont. > 0.3 | 6 / 135 |
> | 7-gram cont. > 0.5 | 3 / 135 |
>
> Both median containment scores are 0.0, indicating that verbatim overlap with public sources is minimal for most items. The small subset with higher overlap (at most 12 out of 135 at the 0.3 threshold) is consistent with our rewriting process, which targets surface-form leakage.
>
> For CTF challenges, the main contamination risk comes from the release of official solutions. We analyze all 85 challenges with official solutions (w/ OS) available online (56.67% of the full set) and compare models' success rates (SR) against the remaining 65 challenges without official solutions (w/o OS). Here are the results:
>
> | Model | SR w/ OS | SR w/o OS |
> |------|------|------|
> | gemini-2.5-pro | 47.06% (40/85) | 67.69% (44/65) |
> | o3-high | 47.06% (40/85) | 63.08% (41/65) |
> | o3 | 42.35% (36/85) | 58.46% (38/65) |
> | o4-mini-high | 38.82% (33/85) | 55.38% (36/65) |
>
> Across all models, the presence of official solutions does not correspond to higher success rates. Models tend to perform worse on problems with official write-ups. This suggests that data contamination has not meaningfully inflated model performance.
>
> **W3:  Number of proof problems.**
>
> Thank you for this valuable comment. We acknowledge that the 30 proof problems are relatively limited in number, which is largely due to the substantial effort required to construct high-quality original questions. As shown in the table in W1, creating and grading the proof component required roughly 90 hours of work from domain experts and PhD students. Despite the relatively small number, we believe proof problems offer a more valuable lens for examining deeper cryptographic reasoning in LLMs, and expanding this component remains a priority for future work.
>
> **W4: Agent framework.**
>
> Thank you for this observation. We acknowledge that the relatively simple agent framework used in our CTF evaluation may not fully capture the upper bound of model capabilities. As noted in our limitations section, this work focuses primarily on assessing the cryptographic abilities of LLMs, and exploring more sophisticated agent frameworks falls outside the current scope. We agree this is a worthwhile direction and leave it for future work.
>
> Thank you again for your thoughtful comments, and we look forward to addressing any remaining questions or concerns.

---

> > ### Author Rebuttal · Reviewer_q2tR · 2026-04-02
> >
> > Thank you for your detailed rebuttals.
> >
> > All my concerns are addressed, and I will keep my positive score.

---

> > > ### Author Response · Authors · 2026-04-04
> > >
> > > Dear Reviewer q2tR,
> > >
> > > Thank you for your thoughtful and constructive review. We are glad that our responses have addressed your concerns, and we sincerely appreciate your time and effort in evaluating our work.
> > >
> > > Best regards,
> > >
> > > Authors

---

### Official Review · Reviewer_hDLK · 2026-03-13

**Soundness:** 2
**Presentation:** 3
**Significance:** 2
**Originality:** 2
**Overall Recommendation:** 2
**Confidence:** 4

**Summary:**

This paper proposes AICrypto, the first comprehensive benchmark specifically designed to evaluate the cryptographic capabilities of LLMs. The benchmark comprises three categories of tasks: 135 multiple-choice questions assessing knowledge memorization, 150 CTF challenges assessing vulnerability exploitation and practical capabilities, and 30 proof problems assessing formal reasoning. Through evaluation of 17 leading models, the authors find that current LLMs have approached or even surpassed human experts in cryptographic concept memorization and routine proofs; however, they still exhibit significant gaps—approximately a 25–30% performance gap—in CTF tasks requiring complex mathematical reasoning, dynamic analysis, and multi-step planning. Furthermore, the authors developed an automated scoring system, achieving a Pearson correlation coefficient exceeding 0.90 with human expert ratings.

**Compliance With Llm Reviewing Policy:**

Affirmed.

**Final Justification:**

The authors have partially addressed my concerns, but the core issues remain (i.e., the drawbacks of using LLM-as-a-judge and benchmark saturation); therefore, I will keep my score unchanged.

**Key Questions For Authors:**

- In mathematics, proof problems are typically solved using tools like Lean. Could similar ideas be applied here—integrating formal verification tools to enable automated proof checking rather than manual grading or llm-as-judge?

**Limitations:**

yes

**Strengths And Weaknesses:**

Strengths:
- Strong domain specificity filling a critical gap: Existing LLM security evaluations predominantly focus on general cybersecurity or generic CTF challenges, lacking systematic benchmarks specifically dedicated to cryptography spanning from theory to practice. AICrypto encompasses the complete spectrum—from foundational concepts and vulnerability exploitation to formal proofs—carrying significant research value.
- Thorough validation of automated scoring: The authors not only proposed employing strong models as graders but also verified high correlation with human expert ratings through 307 samples (Pearson 0.9025), providing credible evidence for the automated evaluation of open-ended problems.
- The paper is well-written and easy to follow.

Weaknesses:
- The approach of using LLMs as judges suffers from significant limitations. For instance, different models—or even distinct versions within the same model family—may exhibit substantial discrepancies when evaluating a single solution, rendering the benchmark assessment less comprehensive and convincing. Furthermore, current models capable of long chain-of-thought reasoning may generate extremely lengthy inference processes, which could exacerbate issues such as "length bias" and "lost in the middle."
- As stated in Section 6 of the paper: MCQ tasks are approaching saturation with limited discriminative power. This renders this section almost incapable of distinguishing among current models and fails to reflect the true reasoning depth of models in cryptography. Consequently, this portion of the benchmark appears to be of limited significance.
- Although 90% of CTF challenges are sourced from 2023 onwards, the training data cutoff dates for some evaluated models (e.g., o3, Gemini 2.5 Pro) potentially extend to mid-2024. Have the authors examined performance disparities for these models on CTF challenges released before versus after their respective training cutoffs? Additionally, are there more stringent decontamination measures implemented (such as n-gram overlap detection or similarity-based filtering)?

---

> ### Author Rebuttal · Authors · 2026-03-27
>
> We really appreciate your time and insightful comments!  We hope the following experiments and discussions can address your questions and concerns.
>
> **W1: LLM-as-Judge limitations.**
>
> Thank you for your valuable comment. To address this concern, we adopt the following strategy described in Section 3.4. When applying LLM-as-Judge for scoring, each model is provided with the question, a reference answer, the scoring criteria, and the answer to be rated. Rather than relying on a single model's output, we employ two strong models, gemini-3.0-pro-preview and gpt-5.1, each scoring independently three times, and taking the average of all six scores as the final result. This ensemble approach reduces the effect of randomness and improves scoring reliability. More importantly, we have compared our automated evaluation against human expert scores. The Pearson correlation between human expert and LLM scores is 0.9025, and the Spearman correlation is 0.8973. These strong correlations suggest that our automatic grading strategy is effective and reasonably reliable.
>
> **W2: MCQ performance saturation.**
>
> Thank you for raising this point, and we have also acknowledged it in Section 6. We respectfully note, however, that the saturation observed in MCQ performance is itself a meaningful finding. The MCQ component is designed to assess recall of fundamental cryptographic concepts, and the high accuracy achieved by current LLMs indicates that they have developed a solid grasp of foundational knowledge in this domain. Since prior work has not systematically examined LLM performance in cryptography, even this "saturated" result provides a useful baseline and a meaningful reference point for future LLM applications in cryptography.
>
> **W3: Data contamination.**
>
> Thank you for raising this important point.
>
> **CTF Challenges.** Since our problems are collected primarily from official competitions, the main source of potential data leakage is the release of official solutions. Community blogs occasionally publish write-ups as well, but these are often inconsistent in quality and frequently contain errors or incomplete reasoning. Below, we show that the presence of official solutions does not lead to higher model success rates in our evaluation.
>
> We analyze all CTF challenges for which official online solutions are available. This accounts for 85 challenges, or 56.67% of the full set. We then compare models' success rates (SR) between problems with official solutions (w/ OS) and those without (w/o OS). Here are the results (Full results are available at https://anonymous.4open.science/r/aicrypto-CE6E/rbt/hDLK.md due to space limits.):
>
> | Model | SR w/ OS | SR w/o OS |
> |------|------------------|---------------------|
> | gemini-2.5-pro | 47.06% (40/85) | 67.69% (44/65) |
> | o3-high | 47.06% (40/85) | 63.08% (41/65) |
> | o3 | 42.35% (36/85) | 58.46% (38/65) |
>
> As shown in the table above, the presence of official solutions does not correspond to higher model success rates. In fact, models tend to perform worse on problems that have official write-ups available. This suggests that data contamination has not meaningfully inflated model performance.
>
> **MCQs.**
> We conduct a retrieval-based contamination probe across all 135 MCQs. For each question, we retrieve the top-10 public candidates using the question text with DuckDuckGo HTML search, then measure normalized n-gram overlap between each MCQ and its nearest retrieved source. The results are as follows:
>
> | Metric | Value |
> | --- | ---: |
> | Median 5-gram cont. | 0.0000 |
> | Mean 5-gram cont. | 0.0722 |
> | Median 7-gram cont. | 0.0000 |
> | Mean 7-gram cont. | 0.0368 |
> | 5-gram cont. > 0.3 | 12 / 135 |
> | 5-gram cont. > 0.5 | 4 / 135 |
> | 7-gram cont. > 0.3 | 6 / 135 |
> | 7-gram cont. > 0.5 | 3 / 135 |
>
> Both median containment scores are 0.0, indicating that verbatim overlap with public sources is minimal for most items. The small subset showing higher overlap (at most 12 out of 135 items at the 0.3 threshold) is consistent with our rewriting and value-modification process, which primarily targets surface-form leakage. We will incorporate a discussion of these aspects into the revised paper.
>
> **Q1: Formal verification tools.**
>
> Thank you for this suggestion. While formal verification tools such as Lean provide strong guarantees, they are difficult to apply here for two reasons. First, our proof problems are expressed in natural language and involve probabilistic reasoning, adversarial models, and high-level proof strategies, making formalization costly, and the current library support is limited. Second, many tasks are open-ended and admit multiple valid or partially correct arguments, whereas formal verification typically yields binary signals for fully specified theorems. Therefore, LLM-as-Judge offers a more practical and flexible evaluation mechanism for our setting.
>
> Thank you again for your thoughtful comments, and we look forward to addressing any remaining concerns.

---

> > ### Author Rebuttal · Reviewer_hDLK · 2026-04-03
> >
> > First and foremost, I sincerely appreciate the authors' clarification. Nevertheless, their rebuttal has not fully allayed my concerns.
> >
> > **Regarding Weakness 1**: I maintain that employing LLMs as benchmark judges is fundamentally problematic. Benchmarks constructed this way cannot be considered truly robust. Esteemed benchmarks such as AIME for mathematics and SWE-Bench Verified for coding rely on deterministic, objective evaluation criteria—only through such rigor can we genuinely characterize model performance. While I acknowledge the authors' use of ensemble methods to enhance reliability, a critical vulnerability persists: the evaluator models themselves (Gemini 3.0 Pro Preview and GPT-5.1) may exhibit instability. Identical prompts can generate inconsistent judgments influenced by positional bias, context length variations, and sampling randomness. Moreover, LLM-as-a-judge methodologies carry inherent risks of being adversarially exploited or hacked.
> > Compounding these issues, frontier models evolve at an unprecedented pace: Gemini has already advanced to 3.1 Pro and GPT to 5.4, while the authors rely on now-outdated versions. This raises acute longevity concerns: what becomes of the benchmark if vendors deprecate these models or discontinue API access? Such vendor lock-in fundamentally compromises the benchmark's sustainability and capacity for broad recognition.
> >
> > **Regarding Weakness 2**: I remain unconvinced that score saturation is acceptable. Benchmarks are designed precisely to differentiate model capabilities; tasks must therefore exhibit adequate discriminative power to fulfill this function. If performance converges near ceiling effects across all tested models, the benchmark forfeits its primary utility and scientific validity.
> >
> > Therefore, I will keep my rating unchanged.

---

> > > ### Author Response · Authors · 2026-04-04
> > >
> > > Thank you again for your thoughtful feedback. We sincerely appreciate your concerns and provide further clarification below.
> > >
> > > # Weakness 1: LLMs as benchmark judges
> > >
> > > **LLM-as-Judge.**
> > >
> > > We do not rely on a single model or a single pass. Instead, we adopt multiple best practices to improve robustness: (i) multi-model ensembling, (ii) repeated scoring, and (iii) validation against human experts. Specifically, we collect 307 human expert annotations and compare them with LLM-based scores, achieving a Pearson correlation of 0.9025 and a Spearman correlation of 0.8973. These strong correlations indicate that, while not perfect, our evaluation protocol is sufficiently reliable for comparative benchmarking in this setting.
> > >
> > > We agree that LLM-as-Judge is not a perfect substitute for fully deterministic evaluation. However, for open-ended proof tasks, where multiple valid reasoning paths and partially correct solutions naturally exist, fully deterministic and objective grading is often infeasible. In such settings, LLM-as-Judge has become a widely adopted practice in the research community [1][2].
> > >
> > > **Longevity concerns.**
> > >
> > > Our evaluation protocol is model-agnostic, i.e., it does not depend on any specific model. Each evaluation instance includes the problem statement, a human-designed reference solution, and explicit scoring criteria. The LLM acts only as an executor of these criteria rather than the source of evaluation standards. As a result, newer or stronger models can be directly integrated into the same protocol without modifying the benchmark design, ensuring long-term usability and consistency.
> > >
> > > **Now-outdated versions.**
> > >
> > > The auto grader models (gpt-5.1 and gemini-3.0-pro-preview) reflect the state of the art at the submission deadline (Jan 28, 2026), which is standard practice. Importantly, our framework is not tied to specific model versions. To further address concerns about evaluator instability and model dependence, we conduct additional experiments using newer frontier models (gpt-5.4 and gemini-3.1-pro-preview), exploring prompt variations and combinations of different models. The results are summarized below:
> > >
> > > | Model | Prompt | Pearson | Spearman
> > > |------|-------|-------|-------|
> > > | gpt-5.1 + gemini-3-pro-preview | original | 0.9025 |0.8974
> > > | gpt-5.4 + gemini-3.1-pro-preview| original| 0. 9137 | 0.919|
> > > | gpt-5.1 + gpt-5.4 | original| 0.8942 |0.8821 |
> > > | gpt-5.1 + gpt-5.4 | variant-1 | 0.8846 | 0.877 |
> > > |gpt-5.1 + gpt-5.4 |variant-2 |0.8994|0.8891|
> > >
> > > Across different model combinations and prompt variants, the correlations with human expert judgments remain consistently high (all Pearson ≥ 0.88 and Spearman ≥ 0.87). Notably, newer models (gpt-5.4 and gemini-3.1-pro-preview) further improve alignment with human scores.
> > >
> > > These results demonstrate that our evaluation protocol is stable across model choices and prompt variations, and does not rely on a specific model version or prompt design. This provides strong evidence that the proposed LLM-as-Judge framework is robust and practically reliable.
> > >
> > >
> > > # Weakness 2: MCQ saturation
> > >
> > > We respectfully clarify that our benchmark evaluates three aspects of cryptographic capability: factual knowledge (MCQs), practical exploitation skills (CTF challenges), and formal reasoning (proof problems). Each component targets a distinct dimension. And we have devoted considerable efforts to every task:
> > >
> > > | Benchmark Part | Task | # Experts | Time (Hours) |
> > > |---------------|------|----------|--------------|
> > > | **MCQs** | Collection, rephrasing, and verification | 3 | 120 |
> > > |  | Human baseline evaluation | 3 | 36 |
> > > | **CTF** | Collection (challenges and solutions), verification | 3 | 300 |
> > > | **Proof** | Proof problem creation | 1 expert + TAs | 50 |
> > > |  | Scoring criteria design, reference solutions, and grading LLM answers | 4 | 40 |
> > > | **Total** | -- | -- | **546** |
> > >
> > > Given the lack of prior systematic evaluation in this domain, it is itself an informative finding. The high performance observed on MCQs indicates that current LLMs have developed a strong mastery of foundational cryptographic knowledge.  At the same time, the CTF and proof components remain clearly discriminative, revealing substantial performance gaps across models.
> > >
> > > Therefore, we believe that our benchmark as a whole continues to provide meaningful and multi-dimensional differentiation of model capabilities, rather than relying on any single component.
> > >
> > > Thank you again for your thoughtful comments, and we look forward to addressing any remaining concerns.
> > >
> > > [1] Gu, Jiawei, et al. "A Survey on LLM-as-a-judge." The Innovation (2024).
> > >
> > > [2] Luong, Minh-Thang, et al. "Towards robust mathematical reasoning." Proceedings of the 2025 Conference on Empirical Methods in Natural Language Processing. 2025.

---

### Decision · Program_Chairs · 2026-04-30

**Decision:**

Accept (regular)

**Comment:**

This work proposes AICrypto, a large comprehensive benchmark for evaluation cybersecurity capabilities of LLMs. This work contains many problems across MCQ, Proof, and CTF, with llm grading correlated against human graders. Proof/CTF show headroom against human performance.

Overall, there is clear agreement among majority of reviewers that this work should be accepted. There is headroom on 2 main areas and thorough analysis, as well as clear methodology and an important area.

There remains a clear weakness: MCQ is already saturated, and as a benchmark the main novelty lies in the remaining unsaturated portion. The findings from this provide some useful analysis.